# Photochemically induced cyclic morphological dynamics via degradation of autonomously produced, self-assembled polymer vesicles

Chenyu Lin [ID] [1], Sai Krishna Katla [ID] [1] & Juan Pérez-Mercader [ID] [1,2 ✉]

Autonomous and out-of-equilibrium vesicles synthesised from small molecules in a homogeneous aqueous medium are an emerging class of dynamically self-assembled systems with considerable potential for engineering natural life mimics. Here we report on the physicochemical mechanism behind a dynamic morphological evolution process through which self-assembled polymeric structures autonomously booted from a homogeneous mixture, evolve from micelles to giant vesicles accompanied by periodic growth and implosion cycles when exposed to oxygen under light irradiation. The system however formed nano-objects or gelation under poor oxygen conditions or when heated. We determined the cause to be photoinduced chemical degradation within hydrated polymer cores inducing osmotic water influx and the subsequent morphological dynamics. The process also led to an increase in the population of polymeric objects through system self-replication. This study offers a new path toward the design of chemically self-assembled systems and their potential application in autonomous material artificial simulation of living systems.

[1] Department of Earth and Planetary Sciences, Harvard University, Cambridge, MA, United States. [2] The Santa Fe Institute, Santa Fe, NM, United States. ✉email: jperezmercader@fas.harvard.edu

iving systems (LSs) (i) process information[1], (ii) metabolize[2,3], (iii) self-reproduce[4], and (iv) evolve[5]. They exist in a fluctuating environment to which they adapt and are thermodynamically open systems working at both the individual and population levels. As time passes their molecular parts degrade; a situation that life resolves through the eventual self-reproduction of LSs. These system-level properties (or patterns) are implemented in our planet with biochemistry's large and complex macromolecules which form (self-organized and self-assembling) permeable membranes, within which operates a complex metabolic network as well as information carrying polymers and all their information handling machinery.

A great deal of progress has been made in the last decades in the search of potential chemical pathways for the generation of the molecular components of some key biochemical macromolecules such as RNA[6–8]. Attempts have also been made to put together systems using extant biochemical materials to emulate in a simplified way some of the above properties (i)–(iv). And there also exist in the literature computer programs which generate objects in computer memory that mimic the plasticity of chemistry and display life-like behavior[9–11].

We can even conceive relatively simple systems of a few stochastic non-linear reaction-diffusion equations[12] which when solved in a computer, exhibit (in the computer screen) properties (i)–(iv)[13]. The fact that constructs displaying the basic patterns of life are far less complex than the simplest forms of extant life, and that they do not make specific references in the computer or in the mathematics to the properties of biochemistry or of any other molecules[1,14,15], inspires us to ask the question: "could one build ex-novo chemical systems which use molecules simpler than biomolecules to express the properties of life?" For example, using small non-biochemical molecules with less complicated bond structures than biomolecules.

Immediately a difficulty presents itself: organic synthesis reactions often require separation, purification and many other separate steps. Hence, the combined yield of the necessary chain of reactions and physical processes to produce certain molecules is very low. This is known as the "Arithmetic Demon" problem[16,17] and is at the base of the so-called "concentration problem" for the origin of life[18,19]. Thus to jointly represent chemically the properties of LSs in an open environment would be practically impossible unless this demon is avoided or somehow controlled. A strategy to achieve this in an ex-novo synthesis suggests itself:[20] identify common physical or chemical features shared by the above properties of life and link the properties through these common features.

Analysis of (i)–(iv) indicates one such common attribute: they depend on the presence of finite free-energy gradients[1–5]. To a first approximation the presence of a (somewhat permeable) membrane enclosing the chemical components of the LS could accomplish this. Furthermore, such a gradient would also facilitate the autonomous boot-up of the chemical system into a functional system with, at least, some of the above properties.

In extant biology the membranes of LSs are quite complex. They are made of phospholipid amphiphiles whose assembled working configurations include vesicles[21] containing all the necessary machinery for life to proceed using chemical fuel provided by its environment.

A permeable vesicle provides a finite free-energy gradient between the interior of the living system and its environment, so that the chemical system is open and remains out of equilibrium and can generate order at a rate congruent with its dimensions and other constrained physico-chemical parameters and variables. The membrane can allow for the necessary exchange of matter and energy between the living system and its environment. Also since energy processing by the living system is limited by its size, it will necessarily decay ("die") in a sufficiently large and varied energy/matter environment[22].

To test the viability for boot-up from a homogeneous mixture using small molecules and subsequently implement in one system the integration of several of the basic properties of life using small molecules, one can formulate the precision (i.e., relatively low polydispersity index) synthesis of polymeric amphiphiles and their dynamic self-assembly into larger than micron-sized vesicles whose physico-chemical evolution in an aqueous medium can be followed in real time. This can be done by the application of RAFT (Reversible Addition Fragmentation Chain Transfer) polymerization to the synthesis of amphiphiles using the techniques of PISA (Polymerization Induced Self Assembly)[23–27]. Originally performed for the autonomous 1-pot, out-of-equilibrium synthesis of amphiphiles leading to vesicles in the hundreds of nanometer size scale in methanol as the solvent medium, it was extended[28] to water as solvent and to generate giant vesicles (GV's are vesicles with diameter larger than 1 um). These vesicles and their spacetime evolution can then be followed "in vivo" using an optical microscope. It was found in reference[29] that this chemical system of small molecules concomitantly leads to a series of emergent behaviors that embody properties associated with natural life. In particular, by using a sequence of alternating photo-illumination pulses the system generates vesicular structures which we called "phoenix", and whose behavior includes growth, collapse and again growth and collapse during several cycles.

In the following we present the results of investigating the chemical causes associated with the "Phoenix" behavior. We will conclude by reporting a pathway to system self-replication associated with chemical degradation. This pathway strongly reminds one of a very primitive form of spore-based reproduction in fungi, ferns, some bacteria and yeast.

## Results

The basis of our experiments will be the RAFT synthesis of amphiphilic block copolymers under PISA conditions in an aqueous medium. This will lead to supramolecular structures emerging from a Dissipative Self-Assembly (DSA) process preceded by the out-of-equilibrium self-organization of the synthesized amphiphilic block copolymers (ABCs). Both the RAFT process and the associated PISA scenario take place under well-controlled reaction conditions. We investigate the impact of oxygen and illumination on the resulting autonomously self-assembled supramolecular structures that are produced.

**Synthesis of PEG-b-PHPMA amphiphiles and resulting morphology.** To study the above, a photo-induced electron transfer polymerization-induced self-assembly (PET-PISA) process was implemented to chain-extend a polyethylene glycol macro-reversible addition-fragmentation chain transfer (m-RAFT) agent with hydroxypropyl methacrylate (HPMA) as monomer and photocatalyzed by a $Ru(bpy)_3^{2+}$ salt[30,31] in an oxygen-poor aqueous medium (oxygen concentration: 0.078–0.017 mM)[32] in a temperature controlled 1.5 mL quartz cuvette that was closed with a Teflon cap to reduce the oxygen entering the cuvette from outside, Fig. 1. After 16 h of blue light irradiation at 25 °C in the reactor, the PISA reaction generated highly defined core–shell nano-structures (degree of polymerization, DP = 20, polydispersity index, PDI = 1.14, average diameters = 10.5 nm, Fig. S1b–d) which have polyethylene glycols as their hydrophilic stabilizers and PHPMA as the core-forming blocks. Using TEM, Fig. S1a, their morphology was characterized as micelles.

After the first 16 h, small aliquots of the resulting reaction solution were transferred to an optical microscope slide for their

**Fig. 1 Light-mediated polymerization-induced self-assembly.** Synthesis route for preparation of micelles via PET-RAFT PISA reaction using PEG-CTA and HPMA catalyzed by Ru(bpy)$_3^{2+}$ under blue light irradiation in an oxygen-poor environment.

observation while the photo PISA reaction continued under illumination by the microscope light. Each of the scenarios we studied will now be presented and the pertinent results discussed.

**Morphological dynamics of an oxygen-poor PISA specimen subject to light irradiation while being observed using optical microscopy.** Prior to the direct optical microscopy observation of the time evolution of the morphological dynamics of these objects in the microscope, small aliquots of the PISA solution were stained with rhodamine 6G (4 uM in the specimen) and then subject to 15 min nitrogen bubbling in order to prepare oxygen-poor microscopic specimens. After this, the aliquots were transferred to a blue plastic frame sealed chamber on a standard microscope glass slide and a cover slip was used to de facto seal the sample on the slide. Then the slide was mounted on the microscope, where the PISA specimens were subject to blue light (470 nm and 6.65 mW power as measured on the slide) irradiation from the microscope light source and in-field fluorescence images were taken every 5 s. In the absence of irradiation, no observable polymer objects were detected in the fluorescence images, even after 16 h, due to their sizes being below the resolution limit of the optical microscope. A fluorescence image taken at 0 min is shown in Fig. 2a. As seen in Fig. 2a and Supplementary movie 1, upon irradiation, a phase rich in the dye, rhodamine 6G, with bright fluorescence emission gradually separated from the water phase and ultimately occupied the entire image field. Given the affinity between the dye and the PHPMA-blocks[33,34], the observed phase separation can be argued to result from gelation which eventually filled the image field with the hydrophobic phase containing PHPMA blocks and HPMA.

This gelation is associated with previously reported temperature-dependent gelation as an inherent property of PEG-b-PHPMA co-polymers[35]. In fact, due to the presence of a large number of hydroxy groups, the core-forming block, PHPMA, in spite of its hydrophobic nature, is a highly hydrated polymer. Chain extension of PHPMA coupled with an increase in temperature produces a limited increase in hydration level in the PHPMA blocks[36] which can swell the self-assembled micelles[37] and/or induce micelle-to-worm evolution[38,39] and eventually result in gelation. (We point out that during irradiation with blue light in the optical microscope, we observed that the environmental temperature near the specimen increased to 37 °C from the initial 25 °C at which the PISA process in the quartz cuvette was carried out.) Therefore, gelation is expected to take place at the blue light irradiated spot of the oxygen-poor specimen due to efficient chain-extension and the subsequent increase in hydration of the polymer cores.

**Morphological dynamics of oxygen-rich PISA specimens subject to light irradiation under the microscope.** If instead of following the procedure described in the previous sub-section, we remove the low-oxygen restriction and we air-bubble the PISA solution (oxygen-rich environment with an oxygen concentration of 0.258 mM)[32] prior to blue light irradiation at the microscope,

once under the microscope, we observe a morphological dynamics which is distinctly different from the one reported in the previous section. The illuminated observed spot in the optical microscope slide became gradually populated by giant polymer objects of various sizes with the simultaneous presence of hollow structures which were characterized as polymer vesicles, Fig. 2b–e, Supplementary movie 2 and Fig. S2. These vesicles were stable when stored in the dark and away from any light. Detailed observation showed that these vesicular structures had either emerged within the field of view of the microscope lens or migrated from outside of the imaged area and gathered towards a spot with the highest light intensity, Fig. S7b. This observed behavior is ascribed to rudimentary phototaxis which is similar to the phenomenon that was reported by Albertsen et al.[29]. Besides the emerging vesicles, in a few of the observed cases, the large nascent micelle aggregates appeared first and were followed by consecutive stages of morphological evolution. First a morphological transition from a micelle aggregate to a vesicle, accompanied by a slight outward budding and internal multi-compartmentalization during the transition. After the onset of vesicle formation, the supramolecular polymer structures started exhibiting what would eventually become cyclic episodes of size-growth accompanied by thinning membranes which, at some maximum sustainable surface area, imploded and became smaller vesicles with proportionately thicker membranes, Fig. 2c. The collapsed vesicles repeated the same process of growth and collapse for a number of times, which in some cases reached about 25 cycles. Interestingly, during such cyclic growth-collapse dynamics, the giant vesicles clearly increased in number (Fig. S3) and gradually filled the entire imaged area (Fig. 2b) of the PISA specimen. (As in reference Albertsen et al. we will refer to this morphological dynamical evolution as "Phoenix" dynamics)[29].

The Phoenix dynamics of course must result from a mechanism different from the previously described gelation observed in oxygen-poor specimens. Indeed it is known[40,41] that in the presence of oxygen, reactive oxygen species (ROS) are generated in PISA systems by the photosensitive species, which in our PISA system includes the photocatalyst, Ru(bpy)$_3^{2+}$, and the staining dye, rhodamine 6G[42]. Therefore, radical polymerization and its contribution to hydrophobic block elongation can be expected to play a limited role in the observed Phoenix dynamics due to its deactivation by the generated ROS.

We know that due to the ongoing polymerization and its many structural and energetic consequences, an increase in temperature (for example due to the reaction or external illumination) is another factor which can potentially promote morphological transitions[43]. Therefore, in order to understand if thermal effects play a role in our observed Phoenix dynamics, an oxygen-rich PISA sample was incubated at 40 °C in darkness. Under these conditions no Phoenix dynamics was observed, although a few micron-sized objects without Phoenix behavior were observed in the fluorescence images, Fig. S4.

In sharp contrast to the above, oxygen-rich samples incubated at 25 °C and exposed to the microscope's blue light

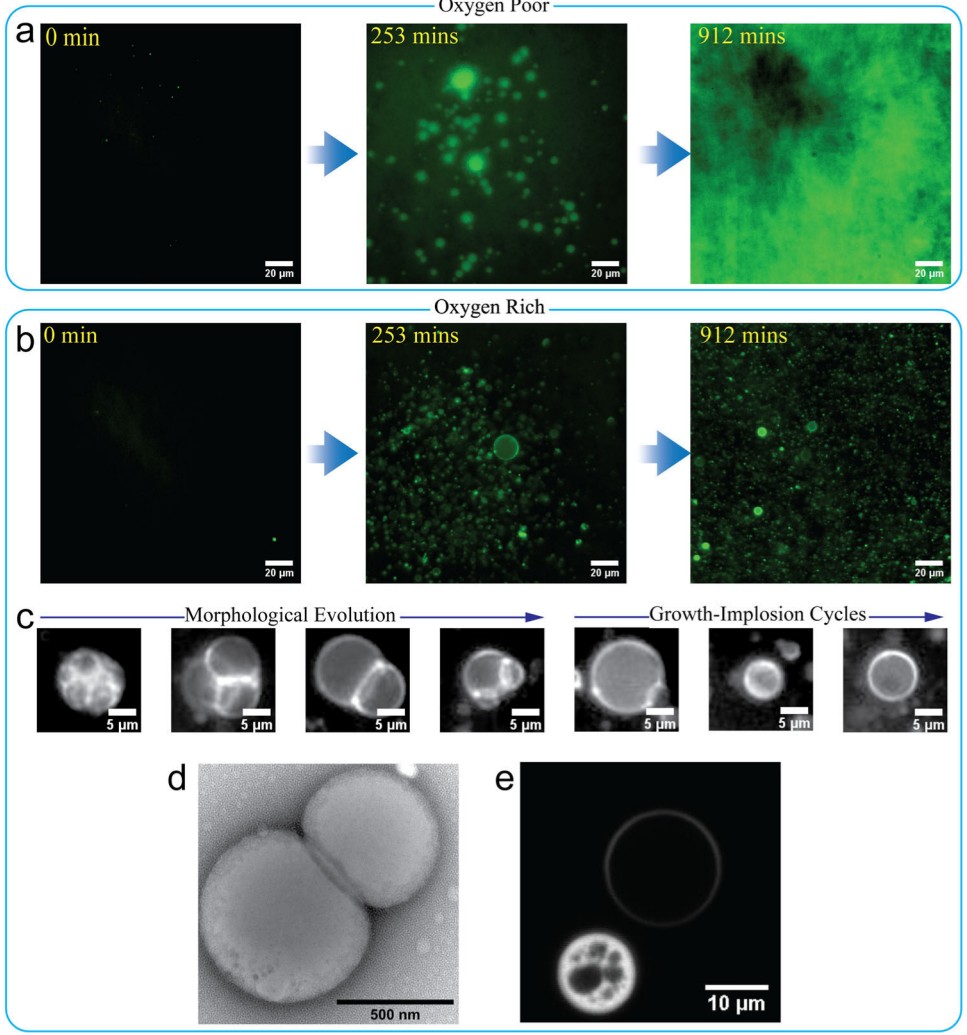

**Fig. 2 Temporal evolution of two distinct morphological dynamics from PISA specimens.** Morphologies of products obtained when exposed to blue light irradiation from the optical microscope. **a** Fluorescence images of the formation of the gelation in an oxygen-poor PISA specimen. **b** Fluorescence images of Phoenix dynamics in the oxygen-rich PISA specimen. **c** Snapshots of two consecutive morphological evolution stages of a large polymer object undergoing Phoenix dynamics in an oxygen-rich PISA specimen. **d** A TEM image of the giant object obtained from oxygen-rich specimen (scale bar = 500 nm). **e** Confocal microscope image of two polymer objects from oxygen-rich specimen after irradiation in a Zeiss Axiovert microscope (scale bar = 10 um).

irradiation (wavelength of 470 nm and 6.65 mW power measured on the slide) exhibited Phoenix dynamics similar to the samples without external temperature control. Therefore, given that the contributions from polymerization and temperature are relatively minor, we infer that the observed experimental behavior indicates that the presence of oxygen plays an important role for Phoenix morphological dynamics to occur when our PISA specimens undergo irradiation in the microscope with the above mentioned blue light. We are then led to interpret the Phoenix dynamics as primarily being the result of water influx into the polymer core of the vesicles driven by osmolarity mismatch between the core of the assembled system and the surrounding solution containing unreacted PISA material. This mismatch originates in an increase of water-soluble species in the polymer core as a consequence of photo-induced oxidative reactions within the self-assembled supramolecular polymer objects. Moreover, under irradiation with blue light in oxygen-rich conditions, the chemical degradation not only produces water-soluble species through photosensitization of the Ru(bpy)$_3^{2+}$ photocatalyst and the R6G dye within the polymeric cores of the

vesicles but also leads to the oxidation of the core forming blocks which results in their increased hydrophilicity[42]. This enhances the osmotic water influx into the vesicles and their subsequent Phoenix dynamics.

**Photo-induced chemical degradation tests**. Our microscope specimens can be considered as closed systems with respect to the transfer of matter. Thus, the oxidative products mentioned above must have originated from the chemicals already present in the PISA solution aliquot that was deposited on the microscope slide. Of course, during controlled radical polymerization, the macro-RAFT agents are the key substances controlling polymer chain extension which, as the reaction proceeds and the degree of polymerization (DP) changes, modifies the packing parameter[21] and leads to a potential sequence of polymer morphologies. It has been reported that many RAFT agents undergo degradation in organic solvents by UV or blue light irradiation[44–47]. For example, in some cases, a nanoscale morphological transition from worms to vesicles can be generated by prolonged exposure to UV irradiation[48].

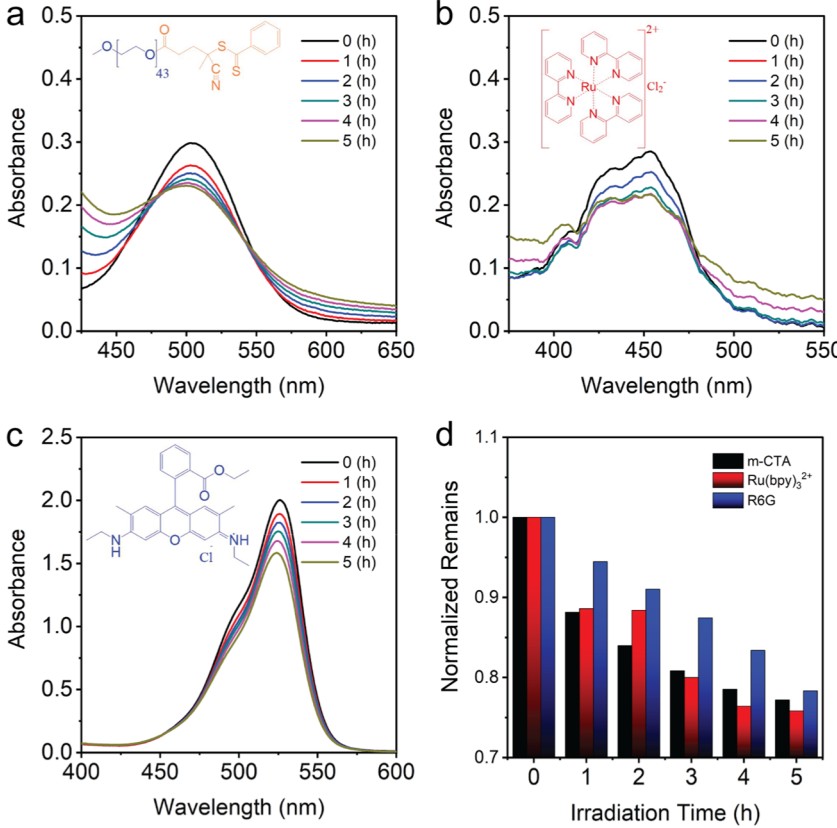

**Fig. 3 UV–VIS spectral analysis.** Photo-induced degradation of three chemicals in oxygen-rich aqueous solutions: **a** Macro-RAFT agent, **b** Ru(bpy)$_3$$^{2+}$, **c** Rhodamine 6 G, **d** Normalized remaining levels of the corresponding photo-degraded chemicals used in PET-PISA solutions.

To examine if our macro-RAFT agent underwent a similar degradation process, we first prepared an oxygen-poor aqueous macro-RAFT solution which was then subject to blue light irradiation and for which we monitored its characteristic absorption peak via UV–VIS spectroscopy. As seen in Fig. S5, an aqueous solution of our macro-RAFT agent exhibits a characteristic absorption with a maximum at 505 nm which corresponds to an n to π* transition[47]. After 5 h of blue light irradiation, the absorbance peak intensity went down by approximately 6%, which indicates a good stability for our macro-RAFT agent in an oxygen poor environment[47].

However, a progressive decrease in the absorbance signature appeared when such photo-induced degradation experiments were performed in oxygen-rich m-RAFT solutions. Up to 23% of the m-RAFT agent undergoing degradation points to the presence of some oxidative reaction taking place due to the presence of oxygen, Fig. 3a. Irradiation of RAFT agents generates carbon-centered radicals. In the presence of oxygen, these radicals can react with oxygen to generate oxidative products including hydroperoxide groups. The irreversible oxidation of parts of living RAFT agents retards polymerization and results in decrease in absorption[32,49]. In addition to m-RAFT agent degradation, Ru(bpy)$_3$$^{2+}$ and rhodamine 6 G (present in our PISA systems) are two photocatalysts well-known to be sensitive to photobleaching[50,51]. Through the reaction with ROS, these photocatalysts degrade and are known to generate aromatic aldehyde, esters[52], hydroperoxide, alcohol, and ketone[53]. As expected, in Fig. 3b, c, we show that both photocatalysts are indeed vulnerable to degradation when irradiated with blue light in an oxygen-rich environment, and show reductions in their absorbance peak intensities of 24 and 22% respectively. Furthermore, the hydrophilic oxidative products formed by the

degradation of m-RAFT, Ru(bpy)$_3$$^{2+}$ and rhodamine 6 G are capable of rapidly dissolving into the water phase and thereby increase the osmotic solute concentrations[32,46,54].

In order to induce osmotic water influx from the surrounding bath into the polymer cores of the vesicles, some chemicals need to degrade within the polymer cores to increase its osmotic solute concentrations. In our control experiments we found that the addition of pre-degraded macro-RAFT agents into an oxygen-poor PISA specimen prior to irradiation with microscope light resulted in gelation instead of Phoenix dynamics which support the above degradation hypothesis. This emphasizes the importance of osmo-larity mismatch induced by ongoing in-core chemical degradation.

**Effects of different monomers on Phoenix dynamics.** Next, we investigated the effects that different monomers can have on Phoenix dynamics. Given a micelle with a highly hydrated core, the oxidative products produced by an on-going in-core chemical degradation tend to dissolve in the nearby internal water-rich domains rather than in the surrounding bath. But the self-assembled micelles in our system consist of amphiphilic diblock copolymers with PHPMA as their hydrophobic blocks. The cores of the micelles contain both HPMA monomer and PHPMA blocks which are in a highly hydrated state due to the presence of a large number of hydroxyl groups capable of capturing water molecules which then induce the formation of many tiny hydrophilic domains within the hydrophobic phase and eventually dissolve the oxidative species. Therefore, it is natural to conclude that the osmolarity mismatch induced by in-core chemical degradation drives a water inflow from the surrounding bath into the polymer core to ultimately induce the hydrophilic

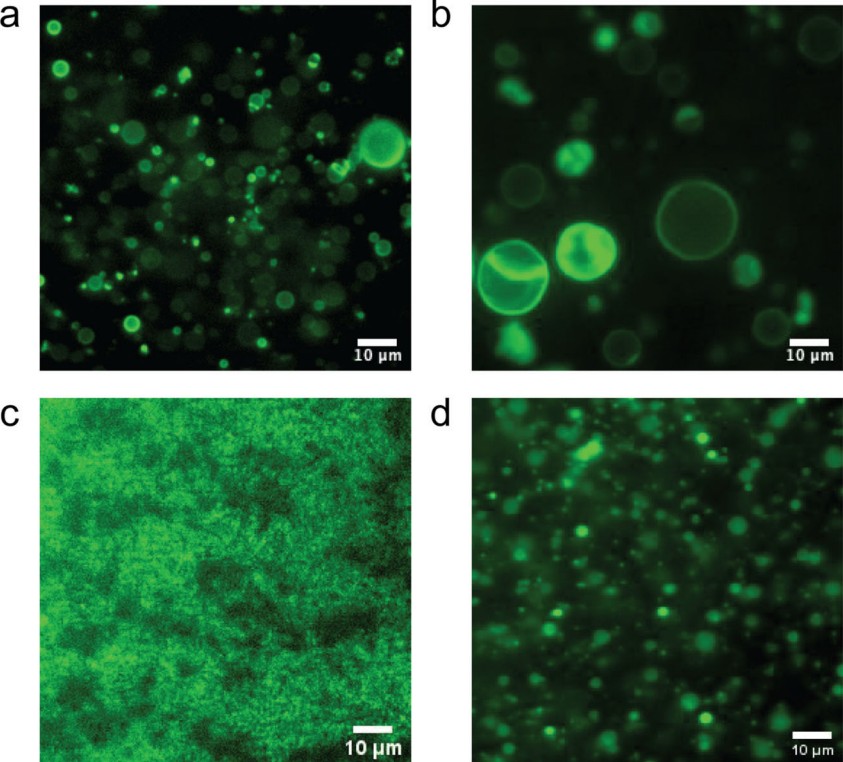

**Fig. 4 Fluorescent images for monomer effects on Phoenix dynamics.** Phoenix dynamics of oxygen-rich PISA specimens when exposed to blue light irradiation from microscope: **a** HPMA, **b** HBA, **c** BA, **d** styrene.

domains to coalesce into a single internal lumen, and lead on to Phoenix dynamics[55].

The relevance of core hydration to Phoenix dynamics becomes more explicit in our PISA experiments (cf. below) in which three other polymers with cores of different hydrophobicities were prepared using different monomers. We used hydroxybutyl acrylate (HBA), butyl acrylate (BA), and styrene as monomers. Of these three monomers, only self-assembled polymer super-structures synthesized using HBA, Fig. 4b, exhibit Phoenix dynamics close to the one with HPMA, Fig. 4a. (Note that the glass transition temperature, $T_g$, for fully dried PHPMA$_{200}$ is 95 °C but $T_g$ for fully hydrated PHPMA$_{200}$ is 47 °C[56]. Higher hydration increases the plasticity of PHPMA[57]. The $T_g$ for the other three monomers is reported to be PBA (−53 °C), PS (100 °C), and PHBA (−40 °C). HBA is a water-soluble monomer with one hydroxyl group just like HPMA. Their similar molecular structure allows their polymer forms to be hydrated to the same extent. With photo-induced degradation occurring in an oxygen-rich environment the polymer cores consisting of PHBA and HBA monomers contain water-rich domains which, as already discussed, can accommodate oxidative products and then experience osmotic water influx. As a result, the resulting self-assembled nanoscale polymer structures evolved and grew in size to form collective micron-scale structures with a mixture of morphologies that included vesicles and vesicle-like objects with outward budding or incipient multi-compartmentalization. We note that the PHBA cores tend to form larger objects than the cores formed with PHPMA, which can be attributed to the higher flexibility of PHBA compared to that of HPMA. The lower $T_g$ of the PHBA molecule allows higher flexibility of the PHBA chains in the cores which can then adapt to a larger lumen expansion by minimizing energy and, eventually, leading to continuous size-growth during hours of irradiation with the light from the microscope.

However, a non-polar monomer such as styrene forms micron-scale emulsions in which the cores have the lowest degree of hydration compared to the other monomers that we studied. Despite the presence of a few micron-scale emulsion droplets, as expected, most self-assembled objects showed only slight swelling and a negligible number of Phoenix dynamics with no observable formation of stable vesicles. On the other hand, BA has a polarity between that of HBA and styrene. It is interesting that with PBA only a few objects exhibited Phoenix dynamics together with the formation of some vesicular structures. However, after hours of irradiation with the microscope's blue light, no large-scale presence of Phoenix dynamics like the ones observed for PHPMA or PHBA cores was observed, although we saw the precipitation of tiny bright objects. This indicates that the majority of the polymer objects tend to grow at nanoscale.

From the results obtained with the use of these selected monomers, we conclude that the hydrated polymer cores are critically important for the presence of Phoenix dynamics. As the chemicals within a polymer-core degrade, the oxidation products disperse into the nearest water phases such as those associated with the hydrated water regions in the core. A water-rich polymer-core consisting of the hydroxyl groups in the monomers and their polymer forms is at a lower energy state, and therefore more stable, than the water-poor polymer cores in capturing and dissolving the oxidation products which is what happens in the larger scale of Phoenix dynamic.

**Application of photoinduced chemical degradation to PET-PISA in reactors.** As already discussed, optical microscopy observations show that Phoenix dynamics takes place and generates giant vesicles when nanoscale polymer objects containing degradable chemicals and water-rich cores are exposed to blue light in the presence of oxygen. We then asked ourselves if by

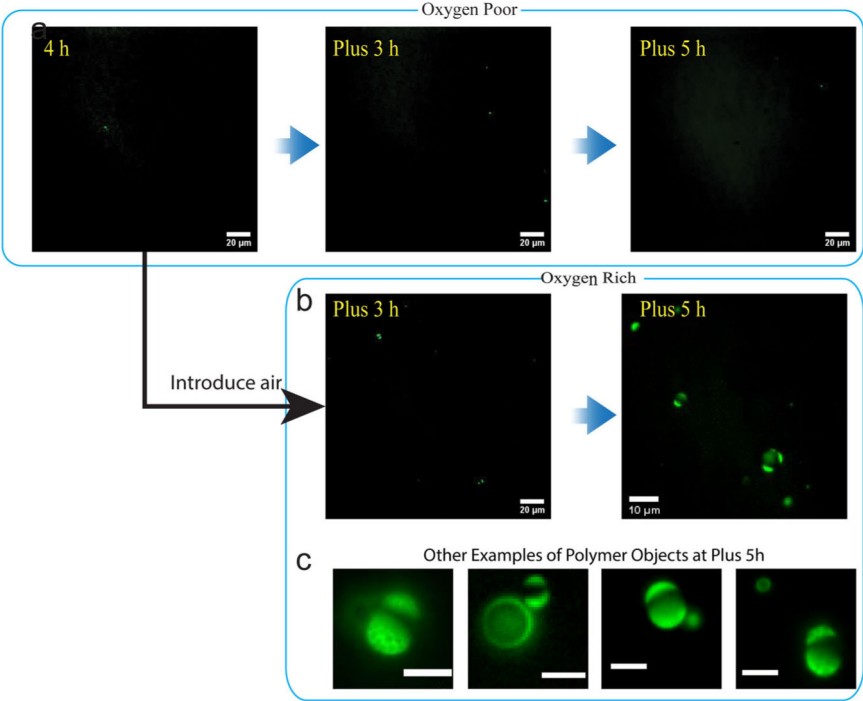

**Fig. 5 Fluorescence images for the effects of in-situ photo-induced degradation.** Fluorescence images of PET-PISA process carried out in oxygen-poor environment: **a** Up to 9 h irradiation showed very few observable objects. **b** After a 4 h oxygen-poor PET-PISA reaction, the reactor was placed under stronger blue light irradiation with introduction of oxygen and rhodamine 6 G. Giant objects with a mixture of morphologies including vesicles and multi-compartmentalized objects emerged during additional 5 h of irradiation time. **c** A few examples of polymer objects at plus 5 h. The scale bars are 5 μm.

applying a similar degradation protocol it would be possible to obtain giant vesicles in a reactor running a conventional (oxygen poor, or run with nitrogen bubbling to remove oxygen) PET-PISA reaction which normally does not generate giant vesicles[58]. To understand whether this would be possible or not, we first conducted a conventional PET-PISA reaction using HPMA as monomer in an oxygen-poor reactor.

As seen in Fig. 5a, most polymer objects reach only nanometer scale sizes and therefore escape detection in the optical fluorescence microscopy imaging even after 10 h of running the PET-PISA process. However, as oxygen (through air bubbling) and rhodamine 6G were introduced into a reactor with an already ongoing conventional PET-PISA reaction for 4 h, we observed that giant objects with hollow or internal multi-compartmental structures gradually form after hours of exposure to a more intense blue light irradiation, Fig. 5b, c. In contrast, such in-situ chemical degradation procedure did not generate similar hollow structures when styrene was used to form the polymer cores. As the results from optical microscopy observations show, the majority of the polymer objects formed with PS cores tend to remain in the form of emulsions and very few or no Phoenix events were observed in experiments performed under these conditions, Fig. S6.

## Discussion

The supramolecular amphiphile block co-polymer structures formed in oxygen-rich PISA conditions showed (Phoenix) dynamics characterized by two consecutive stages of morphological evolution. First, once formed, the micelles evolved to vesicles through intermediate out-of-equilibrium morphologies which include episodes of swelling, budding, and internal multi-compartmentalization. We call this Stage 1 (cf. Fig. 6a). This was followed by a series of cyclic size growth-implosion events which we call Stage 2 (cf. Fig. 6b). Our experiments indicate that

irradiation, oxygen, degradable chemicals, and the hydrated cores of the core–shell self-assembled polymer structures are four key factors responsible for the emergence of this (Phoenix) dynamics. Putting all the above together leads us to the following physico-chemical mechanism and "narrative" to account for the Phoenix dynamics, Fig. 6.

Upon irradiation with blue light in the presence of oxygen, polymer objects containing $Ru(bpy)_3^{2+}$ and rhodamine 6G generate ROS (Reactive Oxygen Species) through photosensitization[42,59,60] which lead to inefficient polymerization (due to radical quenching by the ROS). Compared with this inefficient polymerization, an oxygen-rich environment favors the alternative photo-induced degradation route, which then dominates the PISA system. Species such as $Ru(bpy)_3^{2+}$, rhodamine 6G, and the RAFT molecules at the tails of copolymers contained within polymer cores begin to degrade. Of course, the specific physico-chemical properties of the block copolymers in a given solvent determine the morphology of the self-assembled polymeric structures they can generate. With end-group degradation of the m-RAFT agent, the oxidized tails (degraded parts) of the hydrophobic blocks become hydrophilic which results in affinity towards water domains[46,53,61]. This leads also to a loss of homogeneity in the value of the packing parameter and changes in membrane integrity due to local alterations of the packing parameter in regions of the membrane. In addition, the degradation of the m-RAFT end-groups together with $Ru(bpy)_3^{2+}$ and rhodamine 6G, generate oxidative products which, rapidly, disperse into nearby water domains present within the highly hydrated polymer cores. This builds up the (osmotic) solute concentrations in the cores. Since the permeability of the packing amphiphiles to the oxidative products is negligible when compared with their affinity for water, this difference in concentration between the interior and the exterior of a polymer core generates a dominant osmotic pressure which drives an inflow of water into

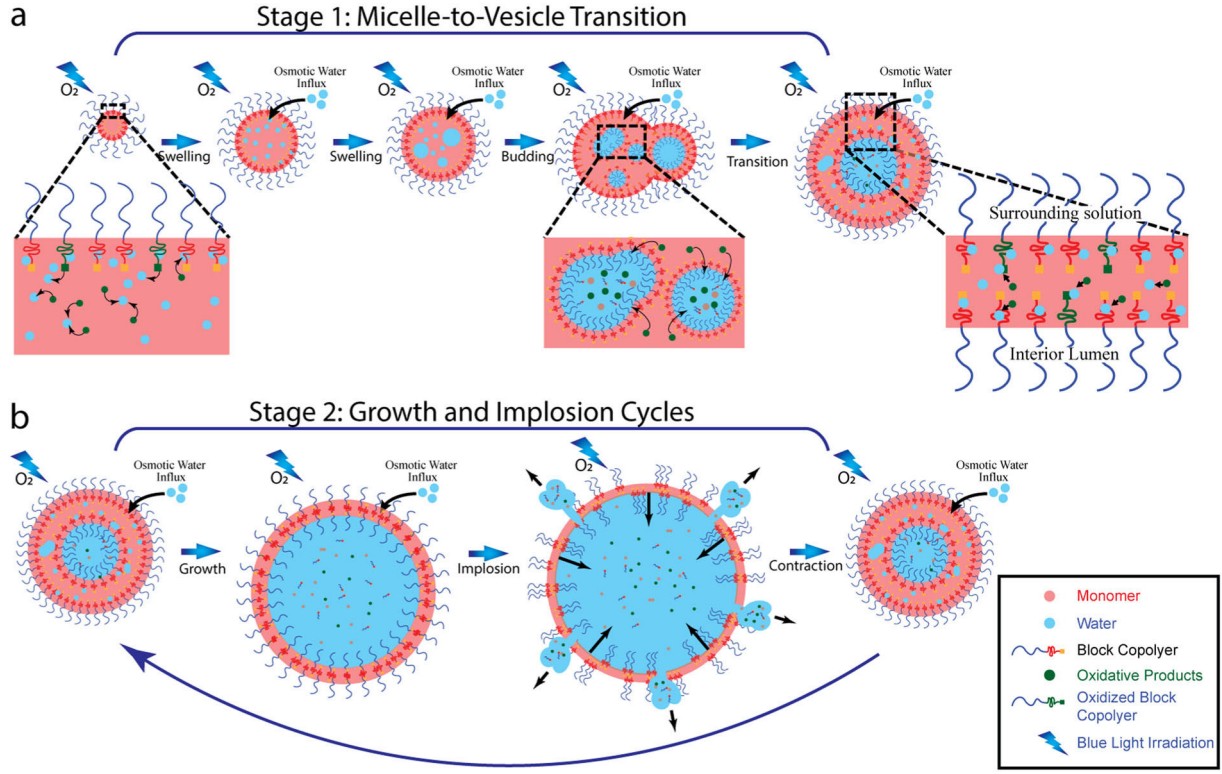

**Fig. 6 Schematic illustration of proposed mechanism for Phoenix dynamics. a** Stage 1: Under exposure of oxygen and blue light irradiation, supramolecular polymer objects transit from micelles to vesicles through a series of intermediate morphologies due to degradation-induced osmotic water influx; **b** Stage 2: The resulting vesicles undergo size growth-implosion cycles.

the core and results in an increase of its degree of hydration. The photo-induced chemical degradation creates a continuous water influx under blue light irradiation, which results in a progressive swelling of the self-assembled polymeric structures from the nanometer to the micrometer scales.

During osmotic water-influx, swelling micelles exhibit outward budding and multi-compartmentalization due to the emergence and subsequent coalescence of progressively growing water compartments that become stabilized by rearrangement of block copolymer molecules with hydrophilic blocks that can wet within the compartment[62]. The latter is associated with the positive correlation between hydration degree and packing parameter. Once the many internal water compartments coalesce to merge into a single large water lumen residing in a polymer core, the containing polymer object acquires a bilayer with the hydrophilic blocks wetting in the aqueous lumen and in the surrounding bath. This morphological transition applies not only to micelles, as micelle aggregates also have hydrated cores and degradable components which can be expected to experience a similar transition[55].

Because of its enriched contents in degradable chemicals, the vesicle bilayer becomes the site for continuous osmotic water influx and allows expansion of the intra-vesicular lumen as well as an increase in the overall size of the vesicle. As expansion of the vesicular volume by continuous photo-induced degradation takes place, more degradable chemicals in the membrane are exposed to irradiation and undergo additional degradation which, eventually, results in a faster (i.e., accelerated) osmotic water influx. However, such outward acceleration is in competition with contraction of the membrane due to its tension, which leads to a variety of time-domain growth modes for the expanding vesicles, Fig. S7a. In our observations, we found that vesicles increased their diameters by up to 2.5-fold which is much larger than a reported 4% expansion range for liposomes

(which did not experience a Phoenix behavior)[54]. This can be explained by the presence of monomers and hydration within the bilayers of vesicles, which act as plasticizers in the membranes and thus provide more flexibility for the expansion of the lumen of the vesicles.

However, the vesicular expansion still has a limit. As a larger lumen expansion results in thinner membranes, and consequently a smaller density of packing copolymers, membrane defects induced by the oxidized copolymer tails become more prevalent and accentuate the weakening of the membrane. As predicted by the Rayleigh–Plesset equation for a "bubble"[63], when the vesicle reaches a critical expansion maximum size the membrane implodes and the inner aqueous solution leaks out to relax the expansion pressure. This relaxation, in turn, allows resealing of the membrane through the pore-line tension[54,64] around the surface defects and results in a smaller vesicle or droplet. The growth in size and implosion episodes proceed in a successive manner as photo-degradation and osmotic imbalance continue.

Remarkably, we observed that Phoenix dynamics always comes accompanied (in due time) by a massive increase in vesicle numbers in the vicinity of the imploding vesicles, Fig. S7b and Supplementary movie 2. The simplest explanation for this observed increase in vesicle number is through a process of vesicle self-replication in which new polymeric objects are formed after each Phoenix cycle due to the increase in available PISA chemicals, including macro-CTA (macro-chain transfer agent) molecules and partially formed amphiphiles, leaked out in the vicinity of vesicles undergoing Phoenix growth-implosion cycles.

In summary, we see that as our thermodynamically open vesicular systems are autonomously booted from a homogeneous mixture, grow in size and age some of their internal components degrade. The individual systems are in a metastable equilibrium state resulting from the confluence of a number of factors, which

include membrane dynamics and tension, osmotic pressure differences between the interior of the system and the field of chemicals in which it is aging, and the chemical degradation of some of the system's key components. This can be thought of as a very primitive form of regulation. Together, the above impel the vesicle membrane to experience an acceleration which modifies the area to volume ratio of the system, which then becomes unstable, collapses, and spills some key chemical components to its environment. Given enough raw materials and stable external conditions (i.e., stability during longer time scales than those in the system's "life-cycle"), the process continues.

## Conclusions

In this paper we have studied a PEG-$b$-PHPMA PISA system which, during blue light irradiation in a microscope used for the observation of the evolution of out-of-equilibrium PISA generated collective structures, exhibited two distinct morphological branches: gelation and an emergent out-of-equilibrium collective dynamics we call "Phoenix dynamics". The highly out-of-equilibrium system avoids biochemistry, boots-up from a homogeneous mixture of small inorganic and carbon-based molecules, is autonomously self-generated, self-assembled, and self-regulated by a combination of physico-chemical properties, such as osmotic imbalances and oxygen-induced degradation.

The presence of oxygen during the PISA process is the key for the system to evolve into these two pathways. Under oxygen-poor conditions, the PET-polymerization occurs efficiently and leads the PISA system to gelation and only the formation of large phase segregated regions. In contrast with the above, in the presence of oxygen-rich (atmospheric proportion) conditions, active photo-induced chemical degradation dominates the PISA system and leads to Phoenix dynamics and the subsequent dissipative self-assembly of easily tracked active giant vesicles. Remarkably, the Phoenix dynamics process opens a pathway for the population growth of the polymer vesicles in the vicinity of the irradiated zone through a unique pathway for system self-replication. This approach to Phoenix dynamics can be extended to other PISA systems where the diblock copolymer has a hydrated hydrophobic core and contains appropriate degradable PISA reagents. These oxygen-dependent photochemical reactions bring new insights and techniques into morphological evolution patterns and dynamical behaviors of giant vesicles and their applications in materials, the ex-novo (or de-novo) synthesis of life-like non-biochemical small molecule systems and the origin of life. We see our results as a relatively simple and completely artificial path in which physical and chemical events can come together in one reaction pot to implement some of the basic properties of living systems.

## Methods

**Materials.** 4-Cyano-4-(phenylcarbonothioylthio)pentanoic acid ($M_w = 279.38$, Sigma-Aldrich), Hydroxypropyl methacrylate (144.17 g/mol, Alfa Aesar), 4-Hydroxybutyl Acrylate (HBA, 144.17 g/mol, >97%, TCI America), Butyl Acrylate (BA, ≥ 99%, 128.17 g/mol, Sigma-Aldrich), Styrene (St, 104.15 g/mol, 99%, ACROS ORGANICS), Rhodamine 6G ($M_w = 479.01$, Sigma), N,N-Dimethylpyridin-4-amine (DMAP, 122.17 g/mol, Aldrich) dicyclohexylcarbodiimide (DCC, 206.33 g/mol, Aldrich), Tris(2,2′-bipyridyl)dichlororuthenium(II) hexahydrate (Ru(bpy)$_3$Cl$_2$ (Mw=748.62, Sigma-Aldrich), Water (HPLC plus, Sigma-Aldrich), Anhydrous Dichloromethane (Sigma-Aldrich), Diethyl ether (Sigma-Aldrich) were used as received. Poly(ethylene glycol) methyl ether ($M_n = 1900$, Alfa Aesar) was dried in a desiccator overnight.

**Synthesis of m-RAFT chain transfer agent.** In a typical synthesis, 1.41 g (Mn = 1900, 0.74 mmole) poly(ethylene glycol) methyl ether, 0.017 g (0.14 mmole) N,N′dimethylaminopyridine, 0.413 g (1.48 mmol) 4-cyano-4-(phenylcarbonoylthio)pentanoic acid, and 10 mL of anhydrous dichloromethane were placed in an oven-dried 25 mL round bottom flask. The mixture was stirred and cooled in an ice bath for 10 min. A mixture of 0.305 g (1.48 mmole) dicyclohexylcarbodiimide in

5 mL of dichloromethane was then added dropwise into the flask. This mixture was then stirred for 24 h. Afterward, the resulting precipitate was filtered and the filtrate was transferred into a large quantity of diethyl ether. The pink-colored m-RAFT chain transfer agent that precipitated in the diethyl ether was later filtered and dried in a desiccator for 2 days before use. The dried product was stored at 4 °C in a refrigerator.

**Preparation of PET-RAFT Polymerization Induced Self-Assembly (PISA) reaction.** At first, 63.44 uL of hydroxypropyl methacrylate, 13.9 mg (6.95 μmol) of Poly(ethylene glycol) methyl ether 4-cyano-4-(phenylcarbonothioylthio) or m-RAFT, 3.26 uL of Ru(bpy)$_3$Cl$_2$ (8.5 mM), and 2 mL water (HPLC Plus, Sigma-Aldrich) were added to a 1 dram glass vial. The mixture was then vortexed for 5 min followed by nitrogen bubbling for 15 min (5 min over the headspace of the liquid and 10 min within the liquid). The mixture was transferred to a 1.5 mL quartz cuvette (purged with nitrogen prior to transfer) which was then capped with a Teflon stopper. The cuvette was then irradiated for 16 h using 7 blue LED units (6.57 mW for each LED unit) and the temperature was maintained at 25 °C using circulation of water around the sample with a pump connected to water-bath. For the other PISA reactions using different monomers such as butyl acrylate, hydroxybutyl acrylate, and polystyrene, equal number of moles of the monomer was used to replace hydroxypropyl methacrylate in the mixture followed by the same reaction process.

**$^1$H Nuclear Magnetic Resonance Measurement ($^1$H-NMR).** 50 uL of PET-RAFT PISA solution was transferred from the reaction cuvette to an Eppendorf tube with 550 uL of methanol-d4 within it. The mixture was vortexed for 5 min and then transferred to an NMR tube. The $^1$H-NMR spectra of the PISA sample was then measured at 25 °C on a 500 MHz Varian Unity/Inova spectrometer.

**Gel permeation chromatography measurement.** After NMR measurement, the solution was transferred from the NMR tube into an Eppendorf tube and speed-dried. The solid compound remaining in the tube was then dissolved in Dimethylformamide with 0.05 mol/L LiBr addition. The mixture was filtered at first by a PTFE syringe filter (pore size is 220 nm) followed by detection using gel permeation chromatography (GPC, Agilent 1260 Infinity II) instrument equipped with a refractive index (RI) detector while eluting with DMF solvent at a flow rate of 1.0 mL/min at 50 °C. Similar procedure was followed for all the PISA samples prepared under different conditions.

**Dynamic Light Scattering (DLS).** 40 uL of PET-RAFT PISA solution was transferred to a disposable polystyrene cuvette and placed in a temperature-controlled chamber in a dynamic light scattering instrument, Malvern Zetasizer Nano ZS. The solution was incubated at 25 °C for 5 min before the measurement of hydrodynamic diameters.

**Fluorescence microscopy and confocal microscopy.** For oxygen-rich samples, an aliquot (72 uL) of PISA solution was transferred to an Eppendorf microcentrifuge tube and stained with Rhodamine 8 G. The mixture was then bubbled with air for 10 min. Afterward, 50 uL of the mixture was transferred to glass microscope slide with a frame-seal slide chamber (15 × 15 mm, 65 μL capacity, BIO-RAD). For oxygen-poor microscopic specimens, the above steps were done under nitrogen purging. The samples were then moved to a fluorescence microscope (Zeiss Axio Observer Z1) and irradiated by blue light ($\lambda = 470$ nm, 6.65 mW). The fluorescence images were captured using green light ($\lambda = 563$ nm) with 50 ms exposure every 5 s. After the irradiation in the Zeiss microscope, the sample was used for confocal microscopy (LSM 880 Confocal Microscope) imaging for further morphological analysis.

**Temperature-controlled fluorescence microscopy.** A sample of PISA solution was incubated at a desired temperature (25 °C or 40 °C) for 30 min in a temperature-controlled incubator (Harvard Apparatus, TC-202A) before further fluorescence microscopy analysis.

**Photoinduced chemical degradation measured by ultraviolet–visible spectroscopy (UV-VIS).** The chemical solutions (i.e., 3.14 mM macro-RAFT, 40 uM rhodamine 6G, or 13.4 uM Ru(bpy)$_3$Cl$_2$ in water (HPLC plus)) to be tested were individually prepared and bubbled with air. For a degradation test, 1 mL of chemical solution was transferred to a temperature-controlled quartz cuvette (temperature = 25 °C) capped with a Teflon stopper. The solution then was exposed to blue light irradiation with 22 blue LED units (6.57 mW for each unit). Every 1 h, aliquots of the solution were measured using UV–VIS spectroscopy (Cole Parmer S2100UV + Spectrophotometer).

**Transmission Electron Microscopy (TEM).** For morphological characterization of the structures formed from PET-RAFT PISA, a small portion of the PISA solution that underwent 16 h of blue LED irradiation was stained with Phosphotungstic Acid and then dropped onto a 400 mesh Copper Grid with Ultrathin Carbon Film (PELCO from

Ted Pella Inc.). After 1 min, the excessive solution was blotted with a filter paper from the grid. The grid was then placed in the dark for overnight drying followed by detection using Hitachi HT7800 electron microscope at a voltage of 80 kV. For characterization of Phoenix dynamic morphology, the portion of the PISA solution that underwent 16 h of blue light irradiation under microscope was extracted using a capillary tube and transferred to the grid. Afterward, the grid was stained by Phosphotungstic acid, blotted with a filter paper, and placed in dark overnight for drying before detection.

## Data availability

Data that support the findings of this study have been provided in the manuscript and the supplementary information. All other relevant data are available from the corresponding author.

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

## Acknowledgements

We thank Repsol, S.A. for their funding of this work and their continued support. We also thank the Harvard Origins of Life Initiative for their support obtaining equipment needed for this work. The funders had no role in study design, data collection and analysis, decision to publish, or preparation of the manuscript.

## Author contributions

C.L. and J.P.-M. conceived and designed the work presented here. C.L. and S.K did the experiments and acquired the data. C.L., S.K. and J.P.-M. did the analysis and interpretation of the data. C.L., S.K. and J.P.-M. drafted and wrote the manuscript. J.P.M. directed and coordinated the study.

## Competing interests

The authors declare no competing interests.
