## [Peer Review File · Communications Chemistry]

Reviewers' comments:

Reviewer #1 (Remarks to the Author):

This is an excellent manuscript from Perez-Mercader's group. The paper reports on the preparation of morphology via a photoPISA approach. The paper is very well organized and the data support the conclusions

I would like to recommend this paper to the editor. I have very minor comments.

1. What means "in an oxygen-poor environment"? It will be better to be quantitative if possible.
2. In the introduction, as the authors used photoPISA, perhaps they could mention this paper: Photoinitiated Polymerization-Induced Self-Assembly (Photo-PISA): New Insights and Opportunities; *Advanced Science* 4 (7), 1700137.
3. The introduction is a bit long in my opinion, and perhaps the authors could cut or condense some sections as this makes the message a bit diffused.
4. I am a bit confused by this sentence: "can be closed to reduce oxygen concentration,". Why close cuvette will reduce oxygen concentration? is it stopping oxygen diffusion? Also, what mean: an oxygen-poor aqueous medium ?
5. The authors should give the oxygen concentration in the samples? The amount of dissolved oxygen in different solvents is provided in this reference: Up in the air: oxygen tolerance in controlled/living radical polymerisation; *Chemical Society Reviews* 47 (12), 4357-4387.
6. line 183, the authors mention: the vesicles gathered towards a spot with the highest light intensity. does this sentence means that the authors see the vesicles moving in this direction or it is an observation that a high vesicle concentration was observed at this spot (due to the high polymerization rate). In my opinion, the former explanation is more plausible.
7. The authors observed a slight decrease of the absorption of RAFT. Perhaps a mechanism could be added.
8. PISA gelation is interesting; can the authors provide a bit more information. Is it due to the formation of worm like particles?
9. The fact there is some degradation of RAFT agents in the presence of oxygen - it will be good to confirm that this does not occur in the absence of oxygen. Another point the authors should consider is the effect of the non-living polymer chains due to the degradation of RAFT agent as noted in Figure 3. This could change the packing parameters in the morphologies.

I think it is a very interesting paper, which deserves publication.

Reviewer #2 (Remarks to the Author):

PISA is currently a hot topic in polymer science. Most of previous research was conducted from the viewpoint of polymer chemistry. This article and other published work performed by the current authors deal with this technology from a different point of view, life science, engineering and polymer physics. This is absolutely of interests to others in the community. It inspires the peers to re-think and re-design the PISA process in a different way. The morphology evolution was monitored by optical microscope (with fluorescence) instead of electron microscopes that are commonly used in the previous reports. Although the accuracy for optical microscope is lower than electron microscope, it is much easier to track the process of morphological changes. The mechanism of phoenix dynamics is impressive and convincing, although similar phenomenon has been reported but interpreted in different ways. The paper is well written and recommended for publish in this journal. The only comment to the authors is to discuss the degradation products of the catalyst and R6G.

Typos: line 90, Page 4: "This can done", "be" is missing.

Reviewer #3 (Remarks to the Author):

Lin and colleagues present a fascinating and well written manuscript demonstrating evolution of micro-sized vesicles from a mixture of nano-sized micelles by virtue of degradation of parts of the forming polymers under irradiation to drive growth and implosion of therein reported vesicles by osmotic pressure difference between the core and surrounding media. It is interesting to watch microstructure come into existence, reminiscent of the appearance of cells on early earth although the chemical species employed in this study are more complex. I thus recommend a minor revision before the acceptance of this manuscript.

-Page 4, line 86

The stricken through text should be deleted

"In order to test the viability for boot-up from a homogeneous mixture using small molecules and subsequently implement in one system the integration of several in of the basic properties of life using small molecules, one can formulate the precision"

-Page 4, line 90

Please insert the article "be"

This can be done by the application of RAFT (Reversible Addition Fragmentation Chain Transfer) polymerization

-The font size of the X-axis in S1b should be increased, its currently illegible yet the caption mentions characteristic chemical shifts.

-Page 7, line 162

What's the reason for the increase in temperature....Does this imply the gelation process was exothermic?? Please expound on this phenomenon.

-Page 9, line 183

Can this be described as some form of rudimentary phototaxis or a diffusion-mediated phenomenon??

-Page 9, line 185

There is no Fig. S7, let alone S7d in the supporting information

-Page 9, line 195

How was the number of vesicles tracked...how did you characterize the increment in the number of vesicles in order to plot Fig S3? By flow cytometry? or by directly counting of vesicles in a microscopy images captured at different times?

-Page 10, line 207

It will be interesting to see what happens at 40 degrees under blue light irradiation....Do you reckon the Phoenix behavior will proceed faster due to increase in kinetic energy of the vesicles??

-Page 10, line 207

How is the stability of the vesicles before implosion when stored?

Reviewers' comments:

Reviewer #1 (Remarks to the Author):

This is an excellent manuscript from Perez-Mercader's group. The paper reports on the preparation of morphology via a photoPISA approach. The paper is very well organized and the data support the conclusions I would like to recommend this paper to the editor. I have very minor comments.

Response: We thank the reviewer for the favorable comments. We have provided below the point-to-point responses to the reviewer's queries.

Question 1. What means "in an oxygen-poor environment"? It will be better to be quantitative if possible.

Response to 1: Oxygen poor environment refers to the samples which were under 15 mins of inert gas (nitrogen) bubbling to minimize the oxygen concentrations. According to the Yeow et al.'s review, the oxygen level of the samples undergoing inert gas bubbling is expected to be around 0.078 to 0.017 mM. We have revised the manuscript to include this oxygen concentration information **on page 5, line 111 of the revised manuscript. And the reference was added as ref. 32.**

Question 2. In the introduction, as the authors used photoPISA, perhaps they could mention this paper: Photoinitiated Polymerization-Induced Self-Assembly (Photo-PISA): New Insights and Opportunities; Advanced Science 4 (7), 1700137.

Response to 2: We thank the reviewer for this suggestion. We have now cited this paper in the introduction of our revised manuscript. **(page 3, line 83 of the revised manuscript as ref. 24)**

Question 3. The introduction is a bit long in my opinion, and perhaps the authors could cut or condense some sections as this makes the message a bit diffused.

Response to 3: **We have revised the introduction and condensed it as much as we could without losing the spirit of what we wanted to convey. We thank the reviewer for their suggestion.**

Question 4. I am a bit confused by this sentence: "can be closed to reduce oxygen concentration,". Why close cuvette will reduce oxygen concentration? is it stopping oxygen diffusion? Also, what mean: an oxygen-poor aqueous medium ?

Response to 4: The solution in the cuvette was prepared by inert gas bubbling and then the cuvette was closed to reduce the intake of oxygen entering from outside. We have changed the sentence to “was closed with a Teflon cap to reduce the oxygen entering the cuvette from outside.” **on page 5, line 112-113 of the revised manuscript.** The oxygen poor aqueous medium refers to the PISA solution in water which is pre-bubbled with inert gas to minimize its oxygen level.

Question 5. The authors should give the oxygen concentration in the samples? The amount of dissolved oxygen in different solvents is provided in this reference: Up in the air: oxygen tolerance in controlled/living radical polymerisation; Chemical Society Reviews 47 (12), 4357-4387.

Response to 5: We thank the reviewer for suggesting this reference. According to it, the oxygen level in our “oxygen-rich environment” is around 0.258 mM and the one with “oxygen-poor environment” is between 0.078 and 0.017 mM. **This information has been included on page 8, line 165-166 (for oxygen rich) and on page 5, line 111 (for oxygen poor).** The reference has also been added to ref. 32 of the resubmitted manuscript.

Question 6. line 183, the authors mention: the vesicles gathered towards a spot with the highest light intensity. does this sentence means that the authors see the vesicles moving in this direction or it is an observation that a high vesicle concentration was observed at this spot (due to the high polymerization rate). In my opinion, the former explanation is more plausible.

Response to 6: We agree with the reviewer’s opinion. We did observe that phenomenon in our time lapse fluorescence imaging. We found that the vesicles regardless of their original locations migrated from outside of the imaging frame and moved in the direction toward the highest light intensity. The vesicles that appeared in the imaging frame showed similar migration tendency.

Question 7. The authors observed a slight decrease of the absorption of RAFT. Perhaps a mechanism could be added.

Response to 7: The mechanism of decrease of the absorption of RAFT can be understood in Li et al.’s research paper and Yeow et al.’s review. In short, irradiation on RAFT agents generates carbon-centered radicals. In the presence of oxygen, these radicals can react with oxygen to generate oxidative products including hydroperoxide groups. The irreversible oxidation of parts of living RAFT agents retards polymerization and results in decrease in absorption. **We have added this mechanism on page 11, line 244-247 and included the references to ref. 32 and ref. 49 of the revised manuscript.**

Question 8. PISA gelation is interesting; can the authors provide a bit more information. Is it due to the formation of worm like particles?

Response to 8: The gelation process is associated with temperature-induced gelation when HPMA is used as monomers. In the absence of oxygen, the local PHPMA chains under irradiation extend for micelles to evolve to worms which gel at higher temperature. In addition, the micelles without evolving to worms can also form gel at higher temperature due to swelling by increasing hydration.

Question 9. The fact there is some degradation of RAFT agents in the presence of oxygen - it will be good to confirm that this does not occur in the absence of oxygen. Another point the authors should consider is the effect of the non-living polymer chains due to the degradation of RAFT agent as noted in Figure 3. This could change the packing parameters in the morphologies.

Response to 9: We thank the reviewer for this comment. We have confirmed the degradation of RAFT agents in the absence of oxygen (oxygen-poor condition) and this information is given in supplementary information, Fig S5. We also agree with the reviewer that the degradation of RAFT agent changes the packing parameters. We had included corresponding discussion in the original manuscript on Page 20, lines 420-424.

Question 10. I think it is a very interesting paper, which deserves publication.

Response to 10: We thank the reviewer for their positive comments.

Reviewer #2 (Remarks to the Author):

PISA is currently a hot topic in polymer science. Most of previous research was conducted from the viewpoint of polymer chemistry. This article and other published work performed by the current authors deal with this technology from a different point of view, life science, engineering and polymer physics. This is absolutely of interests to others in the community. It inspires the peers to re-think and re-design the PISA process in a different way. The morphology evolution was monitored by optical microscope (with fluorescence) instead of electron microscopes that are commonly used in the previous reports. Although the accuracy for optical microscope is lower than electron microscope, it is much easier to track the process of morphological changes. The mechanism of phoenix dynamics is impressive and convincing, although similar phenomenon has been reported but interpreted in different ways. The paper is well written and recommended for publish in this journal. The only comment to the authors is to discuss the degradation products of the catalyst and R6G.

Typos: line 90, Page 4: "This can done", "be" is missing.

Response: We thank the reviewer for their favorable comments about the manuscript. We have corrected the typo in our revised manuscript.

The degradation products formed in the presence of reactive oxygen species (ROS) include aromatic aldehyde, esters, hydroperoxide, alcohol and ketone. We have revised the manuscript to add this discussion in page 11, line 249-250.

Reviewer #3 (Remarks to the Author):

Lin and colleagues present a fascinating and well written manuscript demonstrating evolution of microsized vesicles from a mixture of nanosized micelles by virtue of degradation of parts of the forming polymers under irradiation to drive growth and implosion of therein reported vesicles by osmotic pressure difference between the core and surrounding media. It is interesting to watch microstructure come into existence, reminiscent of the appearance of cells on early earth although the chemical species employed in this study are more complex. I thus recommend a minor revision before the acceptance of this manuscript.

Response: We thank the reviewer for their positive comments about the manuscript. We have now revised the manuscript based on the reviewer's suggestions.

Question 1 -Page 4, line 86

The stricken through text should be deleted "In order to test the viability for boot-up from a homogeneous mixture using small molecules and subsequently implement in one system the integration of several in of the basic properties of life using small molecules, one can formulate the precision"

Response to 1: The stricken through text was probably a formatting error caused while converting the manuscript document to a PDF file. We have checked the manuscript again to remove any stricken through text.

Question 2 -Page 4, line 90

Please insert the article "be"

This can be done by the application of RAFT (Reversible Addition Fragmentation Chain Transfer) polymerization

Response to 2: We have corrected the sentence on page 3, line 81 in the revised manuscript.

Question 3 - The font size of the X-axis in S1b should be increased, its currently illegible yet the caption mentions characteristic chemical shifts.

Response to 3: We have revised Figure S1b and increased the font size of the X-axis.

Question 4 -Page 7, line 162

What's the reason for the increase in temperature....Does this imply the gelation process was exothermic?? Please expound on this phenomenon.

Response to 4: The increase in temperature is due to heat generated from the light source and not due to the exothermic reaction of the gelation process. During irradiation by the microscope light, the electrical components and the lenses gets heated over time. This heat was transferred to the sample during the imaging process, resulting in an increase of temperature from 25 °C to 37 °C at the location of the irradiation. The gelation process is related to this increase in temperature. It is dominated by two collective mechanisms; one is polymerization-induced worm formation which then gel at higher temperature and the other is swollen micelles due to the increase in temperature. **These temperature effects on the gelation is currently part of an on-going study, and the results will be published at a later time.**

Question 5 -Page 9, line 183

Can this be described as some form of rudimentary phototaxis or a diffusion-mediated phenomenon??

Response to 5: From the movies we clearly infer, as mentioned in the Discussion, that there is a photo-induced migration of objects into the field of view, thus indicating the presence of a rudimentary phototaxis, as the reviewer points out. We have added a sentence to the revised manuscript reflecting this reviewer's comment. **(page 8, line 174 of the revised manuscript)**

Question 6 -Page 9, line 185

There is no Fig. S7, let alone S7d in the supporting information -Page 9, line 195 How was the number of vesicles tracked...how did you characterize the increment in the number of vesicles in order to plot Fig S3? By flow cytometry? or by directly counting of vesicles in a microscopy images captured at different times?

Response to 6: When we submitted the manuscript files to the editorial office, the supplementary information file included the Fig. S7. However, there may have been formatting errors during the conversion to a PDF file, which could have caused this issue. Fig S7d is a typo and has been changed to Fig S7b. The increase in the number of vesicles was characterized by analyzing the time-lapse images captured every 5 seconds for 13 hours. The objects in the images were counted using ImageJ software and not by flow cytometry. **This information is added to the legend of Fig. S3 in the revised supplementary information.**

Question 7 -Page 10, line 207

It will be interesting to see what happens at 40 degrees under blue light irradiation....Do you reckon the Phoenix behavior will proceed faster due to increase in kinetic energy of the vesicles??

Response to 7: We agree with the reviewer and their very interesting question.

Temperature/thermal effects are fascinating factors when it comes to the Phoenix dynamics. In addition to the kinetic energy of the vesicles, increasing temperature also increases kinetic energy of molecules including amphiphiles, monomers and photocatalysts. It also accelerates polymerization, increases hydration, and contributes to chemical degradation. The influence of temperature on the Phoenix dynamics is currently part of the on-going study that we are pursuing, and the results will be published at a later time.

Question 8 -Page 10, line 207

How is the stability of the vesicles before implosion when stored?

Response to 8: We thank the reviewer for their question. Indeed, the vesicles were (are) stable when stored in the dark and away from any light. To reflect this, we have added a short sentence to the revised manuscript (page 8, line 170-171 of the revised manuscript).

REVIEWERS' COMMENTS:

Reviewer #1 (Remarks to the Author):

The authors have carefully revised their manuscript and answered all my comments as well as the comments of reviewer #2. I strongly support publication as i think it will attract interest from a broad audience.

Reviewer #3 (Remarks to the Author):

I recommend the acceptance of the current version of this manuscript.